

# Spatiotemporal variation of modern lake, stream, and soil water isotopes in Iceland

David J. Harning[1,2], Jonathan H. Raberg[1,2,3], Jamie M. McFarlin[2], Yarrow Axford[4], Christopher R. Florian[1,2,5], Kristín B. Ólafsdóttir[6], Sebastian Kopf[7], Julio Sepúlveda[1,7], Gifford H. Miller[1,7], Áslaug Geirsdóttir[2]

[1]Institute of Arctic and Alpine Research, University of Colorado Boulder, 80303, USA
[2]Faculty of Earth Sciences, University of Iceland, 101, Iceland
[3]Department of Geology and Geophysics, University of Wyoming, 82071, USA
[4]Department of Earth and Planetary Sciences, Northwestern University, 60208, USA
[5]National Ecological Observatory Network, Battelle, 80301, USA
[6]Icelandic Meteorological Office, 150, Iceland
[7]Department of Geological Sciences, University of Colorado Boulder, 80309, USA

*Correspondence to*: David J. Harning (david.harning@colorado.edu)

**Abstract.** As global warming progresses, changes in high-latitude precipitation are expected to impart long-lasting impacts on earth systems, including glacier mass balance and ecosystem structure. Reconstructing past changes in high-latitude precipitation and hydroclimate from networks of continuous lake records offers one way to improve forecasts of precipitation and precipitation-evaporation balances, but these reconstructions are currently hindered by the incomplete understanding of controls on lake and soil water isotopes. Here, we study the distribution of modern water isotopes in Icelandic lakes, streams, and surface soils collected in 2002, 2003, 2004, 2014, 2019 and 2020 to understand the geographic, geomorphic, and environmental controls on their regional and interannual variability. We find that lake water isotopes in open-basin (through-flowing) lakes reflect local precipitation with biases toward the cold season, particularly in lakes with sub-annual residence times. Closed-basin lakes have water isotope and deuterium excess values consistent with evaporative enrichment. Interannual and seasonal variability of lake-water isotopes at repeatedly sampled sites are consistent with instrumental records of winter snowfall, summer relative humidity, and atmospheric circulation patterns, such as the North Atlantic Oscillation. In contrast to the cold-season bias in Icelandic lakes, summer surface soil water isotopes reflect summer precipitation overprinted by evaporative enrichment that can occur throughout the year, although the soils sampled were shallower than rooting depths for many plant types. This dataset provides new insight into the functionality of water isotopes in Icelandic environments and offers renewed possibilities for optimized site selection and proxy interpretation in future paleohydrological studies on this North Atlantic outpost.



## 1 Introduction

The Arctic hydrological cycle is directly tied to broad earth systems, such as the cryosphere and biosphere, and is expected to change rapidly with continued anthropogenic warming (Serreze et al., 2006). By the end of the century, Arctic precipitation in some regions is forecasted to increase up to 40% – with a progressively larger proportion as rain – due to sea ice retreat, increased local evaporation, and enhanced poleward moisture transport (Bintanja and Selten, 2014; Bintanja and Andry, 2017; Bintanja et al., 2020). As a result, long-lasting impacts to hydrology, glacier mass balance, ocean circulation, ecosystem structure, and global climate system feedbacks are expected (Douville et al., 2021). However, regional-scale predictions of future precipitation changes feature large uncertainties spatially and across models owing to large decadal-scale natural variability and inter-model differences (Bintanja et al., 2020). Developing records that quantify changes in Arctic precipitation beyond the instrumental period in Earth's history provide one means of quantifying how precipitation has changed with past climate variability and can lead to more refined predictions of future precipitation and its impact on other earth systems (e.g., Linderholm et al., 2020; Konecky et al., 2023). Geologic records of past precipitation are, however, similarly limited at present because there are few ubiquitous proxies that clearly relate to specific aspects of hydrologic cycling (Sundqvist et al., 2014).

Lake sediments are a commonly used archive to reconstruct past environmental change due to their wide geographical distribution, sensitivity to local and regional perturbations, and continuous integration of environmental information at the catchment scale (Adrian et al., 2009). In the Arctic, various physical (e.g., grain size, sediment accumulation rates, varves), biological (e.g., pollen, chironomids, and diatoms), and geochemical proxies (e.g., compound-specific stable isotopes) have been used to infer past patterns of precipitation from lake sediment (Linderholm et al., 2020). Of these, the hydrogen or oxygen stable isotopic composition of aquatic and terrestrial macrofossils and lipids are commonly used because they are abundant in lakes and a dominant proportion of the H or O in the preserved structure is derived from plant growth water, which can then be used to estimate past water isotopes through time (Verbruggen et al., 2011; Rach et al., 2017; Holtvoeth et al., 2019). Changes in the past isotopic composition of precipitation, as recorded by water isotope proxies, can therefore yield important information on the response of regional hydrologic cycles to global changes in climate.

However, the stable isotopes signatures preserved in sedimentary archives must be carefully considered with respect to the growth habitat and reservoir of meteoric water that source organisms likely use, changes in the seasonal bias of precipitation, catchment geometry, and changes in landscape dynamics (Sachse et al., 2012; Cluett and Thomas, 2020; McFarlin et al., 2023). Efforts to map the spatiotemporal variability of modern lake water isotopes in highly seasonal climates have advanced our understanding of the processes that control the isotopic composition of lake water on short and long timescales (Gibson et al., 2002; Leng and Anderson, 2003; Jonsson et al., 2006; Cluett and Thomas, 2020; Gorbey et al., 2022; Kjellman et al., 2022). However, such mapping efforts have not yet been conducted in Iceland. Furthermore, while regional studies are not uncommon for lake water isotopes, similar systematic efforts are limited for soil water isotopes, meaning there is a gap in observations for the growth water utilized by terrestrial plants as it isotopically relates to precipitation. Most calibration work for compound-specific water isotope proxies (e.g., *n*-alkyl lipids) use mean annual precipitation (e.g., Sachse



et al., 2012) yet there are local observations that demonstrate Arctic soil water can be seasonally-biased (Cooper et al., 1991; Throckmorton et al., 2016; Bush et al., 2017; McFarlin et al., 2019; Muhic et al., 2023) and/or evaporatively enriched (Tetzlaff

et al., 2018; Eensalu et al., 2023). It is also common to assume that terrestrial plant growth water is restricted to seasonal precipitation because Arctic plant growth is limited to late spring and summer (e.g., Balascio et al., 2013; Curtin et al., 2019; Kjellman et al., 2020; Thomas et al., 2020). These environmental distinctions are important because lake water (aquatic) and precipitation (terrestrial) proxies are often used comparatively to reconstruct hydrologic budgets through time, and differences in the isotopic composition of varying compounds have been used quantitatively to estimate changes in relative humidity (Rach

et al., 2017) or seasonal precipitation amounts (Corcoran et al, 2021; Thomas et al, 2020).

   In this study, we present modern lake, stream, and soil water $\delta^2$H and $\delta^{18}$O data from a network of sites across Iceland (Fig. 1A). We compare lake and soil water isotopes collected during summer and winter field seasons in 2002, 2003, 2004, 2014, 2019 and 2020 with climatological, geomorphic, and geographic parameters to assess which factors may account for the observed isotopic variability. We also present a nearly year-long timeseries of lake outflow water isotopes collected in 2002.

Our data show that in general, lake water isotopes ($\delta^2$H$_{lake}$) in Iceland reflect cold season or annual precipitation with an overprint of evaporative enrichment in closed lake basins. The interannual variability of $\delta^2$H$_{lake}$ is then related to the specific hydrologic budget (i.e., winters with differing amounts of snowfall and/or years of low relative humidity) and atmospheric circulation patterns for individual years. In contrast to lake water, surface soil water isotopes ($\delta^2$H$_{soil}$) broadly demonstrate values that align with summer-biased precipitation and annual evaporative enrichment despite that most precipitation is

delivered in the autumn and winter. Collectively, these data have important implications for the interpretation of water isotope proxies in Icelandic lake sediment records and provide an objective framework within which to estimate past precipitation isotopes in local Holocene archives.

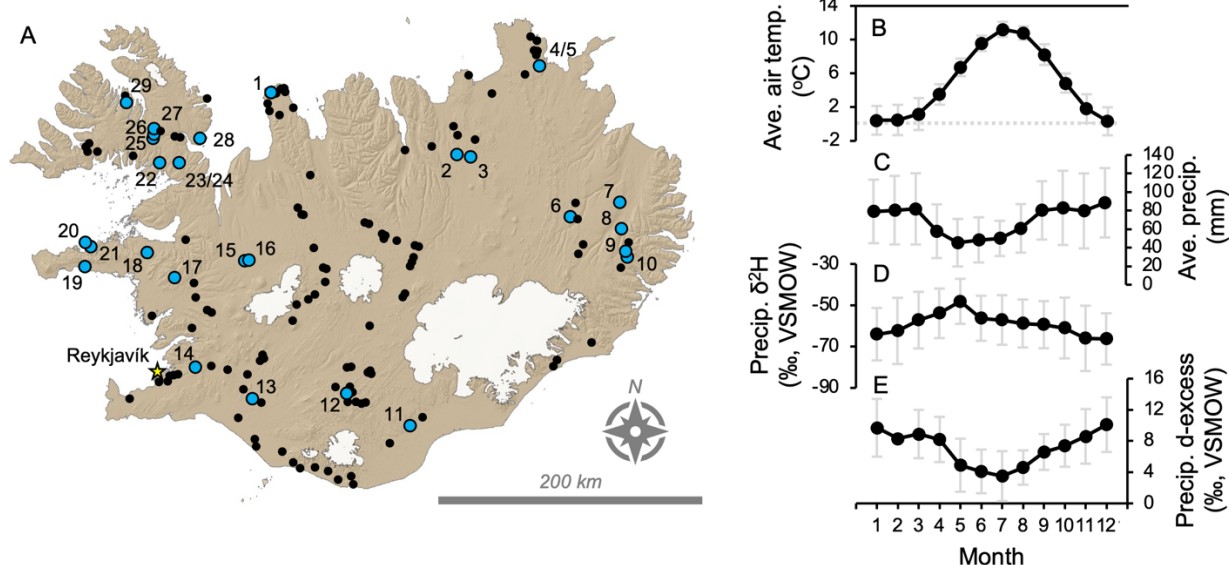

Figure 1: Overview map of Iceland and modern climate. A) Key lake and soil locations from 2014, 2019 and 2020 are shown with
85 blue circles and numerated (see Table 1 for site information), and additional lakes, ponds and streams from 2002, 2003 and 2004 are
shown in black (see Supplemental for site information). B-E) Mean and standard deviation of monthly climate and water isotope
data from Reykjavík, Iceland for the years 1960-1977 and 1992-2018: B) Mean air temperature (ºC) and C) mean precipitation (mm)
from the IMO (IMO, 2023), and D) δ²H (‰, VSMOW) and E) d-excess (‰, VSMOW) from GNIP (IAEA/WMO, 2015).

Table 1: Key sample site information.

| Site # | Site name | Lat (º) | Long (º) | Sample type | Elevation (m) | Residence time (yr) | Hydrologic connectivity | Catchment area (km²) | Lake volume (km³) |
|---|---|---|---|---|---|---|---|---|---|
| 1 | Torfdalsvatn | 66.0605 | -20.3829 | Lake/soil | 52 | 2 | Closed | 2.76 | 0.0016 |
| 2 | Másvatn | 65.6219 | -17.2406 | Lake/soil | 265 | 9 | Open | 19.8 | 0.0427 |
| 3 | Mývatn | 65.5992 | -17.0030 | Soil | 277 | n/a | n/a | n/a | n/a |
| 4 | Stóra Viðarvatn | 66.2369 | -15.8378 | Lake/soil | 151 | 17 | Open | 16.6 | 0.0803 |
| 5 | Litla Viðarvatn | 66.2408 | -15.8064 | Lake/soil | 142 | 0.5 | Open | 1.96 | 0.0004 |
| 6 | Gripdeild | 65.2070 | -15.4454 | Lake | 562 | 2.5 | Open | 17.2 | 0.0091 |
| 7 | Urriðavatn | 65.3168 | -14.4297 | Lake | 49 | 8 | Open | 4.10 | 0.0099 |
| 8 | Skriðdalur | 65.1271 | -14.5381 | Soil | 119 | n/a | n/a | n/a | n/a |
| 9 | Skriðuvatn | 64.9509 | -14.6366 | Lake | 168 | 0.1 | Open | 56.8 | 0.0049 |
| 10 | Heiðarvatn | 64.9023 | -14.5932 | Lake/soil | 442 | 1 | Open | 1.53 | 0.0016 |
| 11 | Systravatn | 63.7901 | -18.0667 | Lake/soil | 120 | 1 | Open | 2.86 | 0.0023 |
| 12 | Frostastaðavatn | 64.0146 | -19.0497 | Lake | 572 | n/a* | Closed | 8.50 | n/a |
| 13 | Vestra Gíslholtsvatn | 63.9414 | -20.5202 | Lake/soil | 61 | 5 | Open | 4.48 | 0.0135 |



| 14 | Leirvogsvatn | 64.2019 | -21.4635 | Lake/soil | 212 | 1 | Open | 26.4 | 0.0121 |
|----|--------------|---------|----------|-----------|-----|-----|--------|------|--------|
| 15 | Efra Grunnavatn | 64.8768 | -20.6719 | Lake | 419 | n/a* | Open | 3.87 | n/a |
| 16 | Ulfsvatn | 64.8878 | -20.5910 | Lake | 440 | n/a* | Open | 13.4 | n/a |
| 17 | Langavatn | 64.7790 | -21.7594 | Lake | 217 | 3 | Open | 81.3 | 0.1303 |
| 18 | Svínavatn | 64.9384 | -22.2372 | Lake/soil | 138 | 0.5 | Open | 5.72 | 0.0017 |
| 19 | Torfavatn | 64.8201 | -23.1642 | Lake/soil | 10 | 0.5 | Open | 3.10 | 0.0011 |
| 20 | Skjaldarvatn | 65.0448 | -22.7919 | Lake/soil | 7 | 0.5 | Closed | 0.64 | 0.0001 |
| 21 | Sauravatn | 65.0149 | -22.7161 | Lake/soil | 8 | 6 | Open | 0.71 | 0.0015 |
| 22 | Berufjarðarvatn | 65.5517 | -22.1053 | Lake/soil | 54 | 0.5 | Open | 3.28 | 0.0003 |
| 23 | Miðheiðarvatn East | 65.5535 | -21.7807 | Lake/soil | 382 | 5 | Open | 0.91 | 0.0018 |
| 24 | Miðheiðarvatn West | 65.5534 | -21.7916 | Lake/soil | 390 | 1.5 | Closed | 0.52 | 0.0003 |
| 25 | Gedduvatn | 65.7159 | -22.1912 | Lake | 470 | n/a* | Open | 3.20 | n/a |
| 26 | Högnavatn | 65.7479 | -22.1709 | Lake | 416 | 0.5 | Open | 2.22 | 0.0004 |
| 27 | Margrétarvatn | 65.7757 | -22.1865 | Lake/soil | 406 | 1 | Open | 1.96 | 0.0006 |
| 28 | Bæjarvötn | 65.7207 | -21.4332 | Lake | 140 | 2 | Open | 10.6 | 0.0100 |
| 29 | Efstadalsvatn | 65.9433 | -22.6668 | Lake/stream | 123 | 0.1 | Open | 35.9 | 0.0014 |

*lake water depth not taken to calculate lake volume and residence time

## 2 Regional Setting

Iceland (~103,000 km$^2$) is situated in the middle of the northern North Atlantic Ocean. Due to its location at the confluence of major oceanic and atmospheric patterns, Iceland's terrestrial climate is sensitive to variations in the strength and position of these systems (e.g., Hanna et al., 2006; Geirsdóttir et al., 2013, 2020). The southern and western coastlines of Iceland are bathed in relatively warm, saline Atlantic waters whereas the northern and occasionally eastern coastlines are influenced by cool and relatively fresh Arctic and Polar waters (Stefánsson, 1962). Iceland is also proximal to the Icelandic Low, a semi-permanent center of low atmospheric pressure that forms one dipole of the North Atlantic Oscillation (NAO), along with Azores High, and is associated with frequent cyclone activity. This pressure difference between the north and south is the dominant mechanism that controls the strength and location of westerly wind and storm tracks across the northern North Atlantic (Hurrell et al., 2003). Given the different oceanographic regimes surrounding Iceland, incoming air masses are strongly influenced by the Atlantic or Arctic waters over which they originate.

The modern climate of Iceland is generally characterized by strong winds, frequent precipitation, mild winters and cool summers (Einarsson, 1984; Hanna et al., 2004). The coldest part of Iceland is the central highlands and snowfall in winter is more common in the north regions. The south coast of the island is warmer, wetter, and windier compared to the northern coast (Hanna et al., 2004, 2006). In Reykjavík, southwest Iceland (Fig. 1A), temperatures average near freezing in the winter months and 10ºC in summer months (Fig. 1B, IMO, 2023). On monthly timescales, precipitation amount in Reykjavík is inversely related to temperature, with a greater proportion of precipitation falling during autumn and winter compared to





summer (Fig. 1C, IMO, 2023). However, on longer-term decadal timescales, precipitation amount shows a moderate positive

correlation with temperature (Hanna et al., 2004). The isotopic composition of precipitation recorded at the Global Network

of Isotopes in Precipitation (GNIP) station in Reykjavík is on average most depleted during the winter months and most

enriched in the month of May, although there is large year-to-year variability (Fig. 1D, IAEA/WMO, 2015). This variability

reflects the competing moisture sources to Iceland and likely explains an a lack of correlation between $\delta^2H$ of precipitation

and air temperature (Fig. S1A). Deuterium excess (d-excess) values of precipitation are generally highest during the winter

and lowest during the summer (Fig. 1E, IAEA/WMO, 2015). Reykjavík's annual Local Meteoric Water Line (LMWL, Fig. 2,

IAEA/WMO, 2015) of:

$$\delta^2H = 7.78 \times \delta^{18}O + 5.29 \tag{1}$$

has a similar slope but lower d-excess than the Global Meteoric Water Line (GMWL, Craig, 1961) of:

$$\delta^2H = 8*\delta^{18}O + 10 \tag{2}$$

LMWLs of the different seasons (winter, DJF; spring, MAM; summer, JJA; and autumn, SON) are variable, where summer

plots distinctly below the annual LMWL and winter, spring, and autumn plot closer to and typically above the LMWL (Fig.

S1B). In hydrologically closed lake surface water systems, kinetic fractionation from evaporation will result in isotopically

enriched water relative to precipitation and low d-excess values (Feng et al., 2016; Bowen et al., 2018). These are more

appropriately described by a Local Evaporation Line (LEL, Fig. 2), which plots below the LMWL (Gat, 1996).

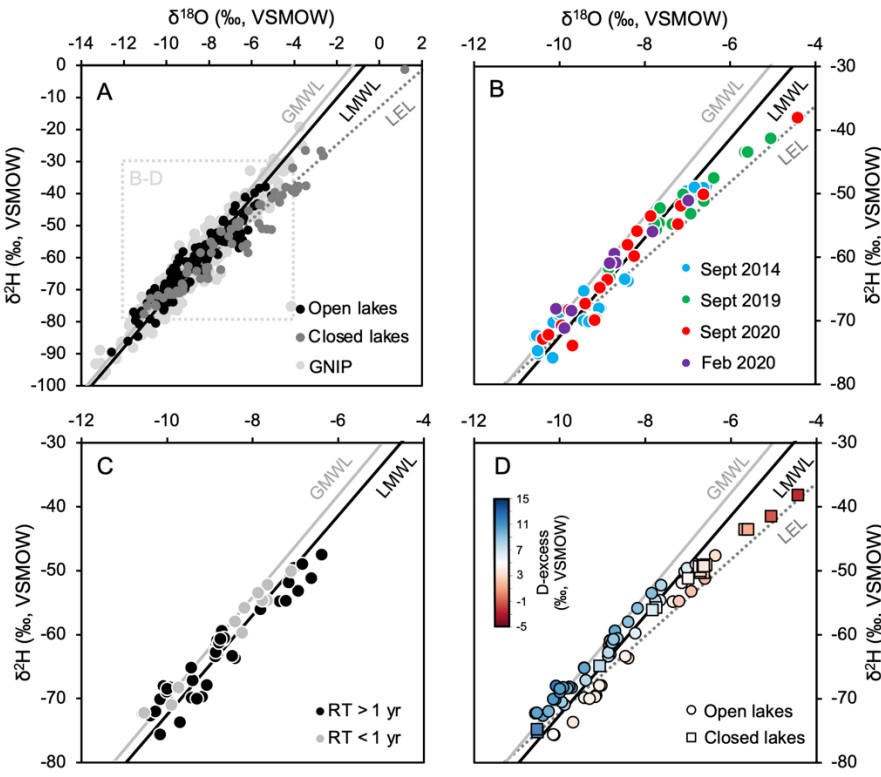





**Figure 2: Lake water stable isotopes from Iceland plotted against the GMWL (Craig, 1961), LMWL (GNIP Reykjavík, IAEA/WMO, 2015), and LEL (closed lakes, this study). Samples are separated by A) hydrologic connectivity for all lakes (open/closed, *n*=223), B) year and month of collection for all lakes (open/closed) from 2014, 2019, and 2020 (*n*=80), C) residence time for open lakes only from 130 2014, 2019, and 2020 (RT, *n*=61), and D) d-excess values for all open/closed lakes only form 2014, 2019, and 2020 (*n*=80).**

The mountainous topography of the island, particularly in the southeast, produces a strong orographic effect on precipitation, with annual precipitation in excess of 5000 mm atop glacial summits, such as Vatnajökull and Mýrdalsjökull (Einarsson, 1984; Crochet et al., 2007). Along with southerly to southeasterly prevailing winds, precipitation amount tends to 135 be heaviest in south and southeast Iceland with a rain shadow over the central highlands, especially north of Vatnajökull (Einarsson, 1984). This geography results in distillation of air masses as they move inland and thus progressively more depleted $\delta^2$H values of precipitation at higher altitudes farther from the coastline (Bödvarsson, 1962; Friedman et al., 1963; Árnason, 1976). In addition, temperature and variations in moisture source also exert strong controls on precipitation isotopes in Iceland (Bödvarsson, 1962; Friedman et al., 1963) and across the North Atlantic (Dansgaard, 1964; Jouzel et al., 2000; Bowen and 140 Revenaugh, 2003; Sodemann, et al., 2008). Moisture sources predominately fluctuate between southerly and northerly directions.

## 3 Methods and Materials

### 3.1 Samples Collection

We present a compilation of water isotope data collected over multiple field campaigns in 2002, 2003, 2004, 2014, 2019, and 2020. Between 2002 and 2004, we collected lake water samples from 135 lakes and ponds (mainly in summers, with several sites revisited in February/March, Fig. 1A), as well as geothermal streams (*n*=2), non-geothermal streams and lake water outflow (*n*=8), and snow meltwater (*n*=3) (Fig. S3). For one lake outflow stream in NW Iceland (Efstadalsvatn), Ragna Aðalsteinsdóttir, the farmer residing at the site, collected water samples every 2 weeks between March and November 2002. 150 We visited a total of 26 lakes in summer 2014 (*n*=14), summer 2019 (*n*=17), winter 2020 (*n*=4), and summer 2020 (*n*=19) (Fig. 1A and Table 1). Of lakes visited during the summer field season in 2014, 7 were revisited in summer 2019 and 2020. All lakes visited during the summer field season in 2019 were also visited during the subsequent summer field season in 2020. During each field season, we sampled surface water from the center of each lake using Nalgene bottles. For select lakes in summer 2019 (*n*=2) and winter 2020 (*n*=4), we also sampled bottom water from the center of the lake using a Niskin water 155 sampler. All water samples were immediately transferred to glass vials with no headspace, sealed with Teflon tape and/or electrical tape to prevent evaporation, and refrigerated prior to isotopic analysis. Further details on sampling locations, additional proxy data, and associated water chemistry for the 2014, 2019 and 2020 datasets can be found in Florian (2016) and Raberg et al. (2021a, 2021b, 2023).

We sampled a total of 23 surface soil samples from 18 locations in summer 2019 (Fig. 1A and Table 1). Of the 18 160 locations, 16 are within the catchments of lakes sampled for lake water during the same field season as described above. The



two remaining soil sites (Skriðdalur and Mývatn) do not have corresponding lake water samples as they were targeted to provide forested endmembers. Soil samples were collected to evenly integrate the top 10 cm of soil. Each soil sample was sampled in 50 mL Falcon tubes, which were fully filled, sealed with Teflon tape, and kept refrigerated to prevent evaporation of water before isotopic analysis. Further details on sampling locations and additional proxy data datasets can be found in

Raberg et al. (2024).

### 3.2 Water isotope analysis

Lake water samples were analyzed in the Stable Isotope Laboratory at the Institute of Arctic and Alpine Research, University of Colorado Boulder. Samples from 2002 to 2004 were measured via mass spectrometry methods using the $CO_2$ equilibration

method for oxygen isotopes (Epstein and Mayeda, 1953) and the uranium reduction method for hydrogen isotopes (Vaughn et al., 1998). For the remainder of the sample set, stable isotopes measurements ($\delta^2H$ and $\delta^{18}O$) were measured using a Picarro L2130-i Cavity Ring-Down Mass Spectrometer. Each sample was run 6 times, with the first three data points discarded to prevent memory isotope effects. Mean values were calculated using the final three measurements. A replicate was run after ~20 samples to check for reproducibility. We report data using $\delta$ notation in per mil (‰) relative to VSMOW:

$$\delta = 1000 \times \left(\frac{R_{sample}}{R_{VSMOW}} - 1\right) \tag{3}$$

where R is $^2H/^1H$ and $^{18}O/^{16}O$. Samples were analyzed against a suite of secondary laboratory standards that have been vigorously calibrated to primary reference materials, VSMOW2 and SLAP. The secondary laboratory standards (Florida water, Boulder water, and Antarctic water) have $\delta^2H/\delta^{18}O$ values of -2.82/-0.76, -111.65/-14.15, and -239.13/-30.30 ‰, and an analytical error of 1 ‰ and 0.1 ‰, respectively.

All soil samples were sent to the University of Utah Stable Isotope Research Facility for soil water vacuum line extraction and isotope measurements (see Cook et al., 2017 for methodology). Values are reported in delta notation relative to VSMOW.

Deuterium excess (d-excess) values were calculated after Dansgaard (1964) as:

d-excess = $\delta^2H$ – 8 x $\delta^{18}O$ (4)

Given that $\delta^2H$ and $\delta^{18}O$ values are generally correlative, we focus our discussion on $\delta^2H$ and d-excess values for simplicity.

### 3.3 Additional site information

For each location, we compiled and computed additional information to test which environmental parameters are related to lake and soil water $\delta^2H$ values.

For all lakes, we determined whether they were open or closed basins via the presence/absence of surface outflow in

summer 2023 satellite imagery, and measured elevation (m) and closest distance from coast (km) in Google Earth. We recognize there may be some ambiguity in this binary open/closed designation as some lakes that are at times open basins can be closed in late summer or in a year of low snowpack, for example. For lakes visited in 2014, 2019, and 2020, water depths





were measured using a weighted measuring tape from the lake depocenter, and lake surface area (km$^2$) and lake catchment area (km$^2$) were measured in Google Earth. A complete model of lake bathymetries is unavailable, so we estimate lake volumes (V, km$^3$) based on half of an ellipsoid:

$$V = \frac{1}{2}(\frac{4}{3} \times A_{lake} \times Z) \tag{5}$$

where $A_{lake}$ is the lake surface area and Z is the maximum depth of the lake. Lake water residence times (RT) were calculated after Jonsson et al. (2009):

$$RT = \frac{Lake\ volume}{Runoff} \tag{6}$$

where

$$Runoff = Catchment\ Area \times Precipitation \times \frac{R}{P} \tag{7}$$

and R/P is the runoff/precip ratio (Gibson and Edwards, 2002), where the latter is less than 1 because some precipitation that falls in a lake catchment is lost to evaporation, transpiration, sublimation, or groundwater flow. Quantitative information on these processes for Icelandic lakes is limited, so we use an R/P value of 0.5, consistent with existing Arctic lake studies elsewhere (Gibson et al., 2002; Gorbey et al., 2022). We acknowledge that the R/P value may vary between sites, and the role of groundwater is an important unknown, but as we are more broadly interested in the correlation of residence time with water stable isotopes, an estimate of 0.5 is likely sufficient to address this variable. Finally, we obtained mean annual air temperature (MAT), mean annual precipitation (MAP), as well as seasonal means from the closest weather station to each lake operated by the Icelandic Meteorological Office (IMO, 2023), including the number of years preceding sample collection equivalent to individual lake residence times. As the elevation of the closest weather station often differs from the elevation of the lake site, we also scaled temperature integrations for each lake according to the global lapse rate (6 ºC/km).

For soil sites, we used Google Earth-derived elevation and distance from coastline information as described above for lake sites. In addition, we measured soil water content (SWC):

$$SWC = \frac{wet\ mass - dry\ mass}{dry\ mass} \times 100 \tag{8}$$

and *in situ* soil temperatures for one year using iButton loggers deployed at 10 cm soil depth in summer 2019 and collected in summer 2020 (Raberg et al., 2021b). iButton temperatures were recorded every 3 hours, which we then used to produce mean annual and seasonally integrated values. We additionally used iButtons to measure *in situ* surface and bottom water temperatures at 6-hour intervals over the same period for four lake sites (Raberg at al., 2021b). Finally, we include information on the dominant vegetation cover above each soil sample that we observed in the field during collection: forest (*Betula pubescens*), dwarf shrub (*Betula nana*), heath shrub (Ericaceae), moss/lichen/prostrate, grass, and moss. Except for Torfavatn, which is classified as a histosol (TOC > 20 %), all collected soils are classified as brown-gleyic andosols (TOC < 12 %) (Arnalds, 2004).





### 3.4 Statistical analyses

We computed Pearson correlation matrices and *p*-values to assess the relationship between water isotopes, d-excess and various
environmental parameters using the open-source software R (R Core Team, 2022) and the corrplot package (Wei and Simko, 2021). To assess the significance of lake and soil water isotope population differences and of water isotopes and environmental variable correlations, we performed student *t*-tests. We define significant results as those with *p*-values < 0.05.

### 4 Results

The $\delta^2$H of lake water ($\delta^2$H$_{lake}$) ranges from -89.53 to -1.35 ‰ and falls within the natural range of meteoric water isotopes recorded at the Reykjavík GNIP station (Fig. 2). Spatially, $\delta^2$H$_{lake}$ values tend to be more depleted in northern Iceland compared to the south (Fig. S2). The d-excess values of lake water range from -11 to 15.71 ‰, where open lakes are typically higher, ranging from -4.30 to 15.71 ‰ (Fig. 2D) and exhibit no clear spatial pattern (Fig. S2). The $\delta^2$H of stream, snowmelt and geothermal spring water values also fall within the natural range of meteoric water isotopes, ranging from -89.55 to -55.66 ‰, 235 -79.11 to -60.13 ‰, and -79.37 to -73.93 ‰, respectively (Fig. S3), and d-excess values range from -6.58 to 14.91 ‰. In general, lake water isotopes for all open lakes plot along the LMWL, whereas water isotopes for closed lakes evolve along a LEL (Fig. 2A). Using data from all closed lakes, the LEL is described by the following relationship:

$$\delta^2H = 5.90 \times \delta^{18}O - 13.09 \ (r = 0.97, p < 0.05) \tag{9}$$

Using data from 2014, 2019, and 2020, open lakes that have residence times of less than one year (efficient flushing) follow
the annual LMWL closely. In contrast, open lakes that have residence times over one year (less efficient flushing) feature larger deviations from the LMWL (Fig. 2C).

For all lakes (open/closed), we find strong correlations between $\delta^2$H$_{lake}$ and mean annual (r = 0.66, *p* < 0.05), mean winter (r = 0.67, *p* < 0.05), mean autumn (r = 0.69, *p* < 0.05), and mean spring temperature (r = 0.64, *p* < 0.05), and a weak but still significant correlation with mean summer temperature (r = 0.38, *p* < 0.05) (Fig. 3A). However, we find no significant 245 correlations between mean annual precipitation amount and $\delta^2$H$_{lake}$ values nor any seasonal precipitation amount (Fig. 3A). If closed lake basins are excluded, we find similar relationships between $\delta^2$H$_{lake}$ and temperature (Fig. 3B). However, excluding closed lakes also reveals that open lakes are weakly correlated with mean spring (r = 0.37, *p* < 0.05) and mean summer precipitation amount (r = 0.33, *p* < 0.05) (Fig. 3B). Using the entire lake dataset, we find a moderate correlation between $\delta^2$H$_{lake}$ values and elevation (r = -0.46, *p* < 0.05) and a weak correlation between $\delta^2$H$_{lake}$ values and distance from coastline (r = -0.33, 250 *p* < 0.05) (Fig. 3). Lake water d-excess values show weak to moderate correlations with most variables in all lakes, including elevation, distance from coast, precipitation amount and temperature (Fig. 3A). However, when closed lakes are excluded, d-excess values are no longer correlated with precipitation amounts, except for mean autumn precipitation (r = -0.44, *p* < 0.05) (Fig. 3B). Summer $\delta^2$H$_{lake}$ values are relatively more enriched in 2019 than summer $\delta^2$H$_{lake}$ values in 2014 and 2020 (Figs. 4A and 5). Summer d-excess values are also relatively lower in 2019 compared to 2014 and 2020 (Fig. 4B and 5). For the 4 lakes





that have seasonal data, winter surface $\delta^2H_{lake}$ values are more depleted than summer surface $\delta^2H_{lake}$ values, with differences ranging from -0.54 to 16.51 % (Fig. 6A). Winter lake water d-excess values are relatively higher than summer d-excess values, except one lake, Stóra Viðarvatn (Fig. 6B). For the 2002 lake outflow stream timeseries from Efstadalsvatn (Fig. 1A), $\delta^2H$ values range from -78.75 ‰ at the end of April to -70.68 ‰ at the end of June and are generally $^2$H-enriched between June and August (Fig. 7A), and d-excess values range from 11.13 to 14.29 ‰ with the highest values obtained in June (Fig. 7B).

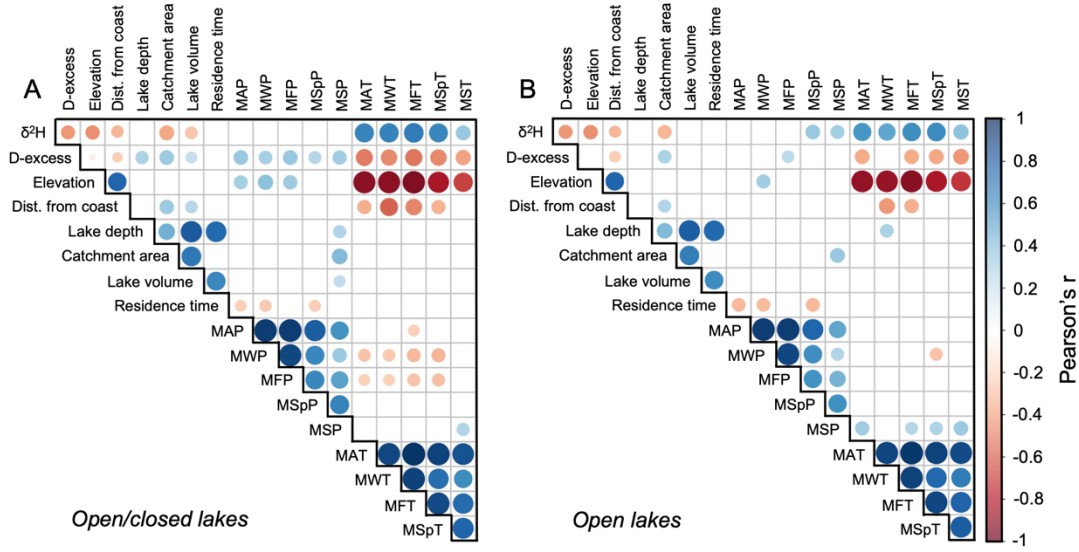


**Figure 3: Pearson correlation matrices for A) all open and closed lakes and B) only open lake water $\delta^2H$ (‰, VSMOW), d-excess, and various climatological, geographic, and environmental parameters. Elevation and distance from coastline ($n$=223); lake depth, catchment area, lake volume ($n$=80); residence time and annual and seasonal precipitation amount ($n$=61); and annual and seasonal temperature ($n$=80). MAP, mean annual precipitation; MWP, mean winter precipitation; MFP, mean autumn precipitation; MSpP,**
**mean spring precipitation; MSP, mean summer precipitation; MAT, mean annual temperature; MWT, mean winter temperature; MFT, mean autumn temperature; MSpT, mean spring temperature; MST, mean summer temperature. Empty boxes reflect statistically insignificant correlations ($p > 0.05$).**





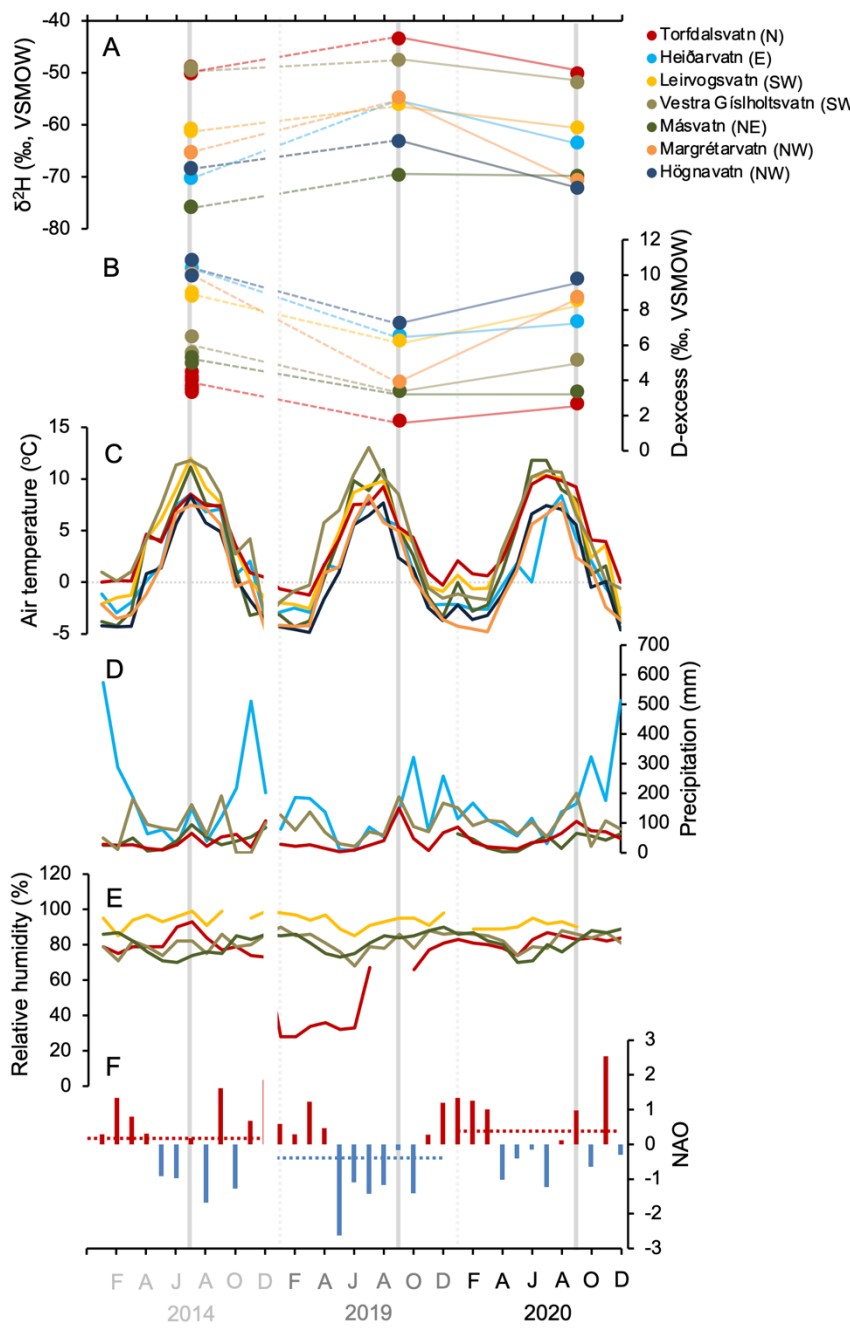

**Figure 4: Interannual variability of lake surface water δ²H compared with climatological datasets. A) δ²H (‰, VSMOW) of surface water from 7 lakes collected in July 2014, February 2019, and September 2020, B) corresponding d-excess values (‰, VSMOW), C) mean monthly air temperature (ºC) for each lake, D) mean monthly precipitation (mm) for 4 lakes (not available for Leirvogsvatn, Margrétarvatn and Högnavatn), E) mean monthly relative humidity (%) for 4 lakes (not available for Heiðarvatn, Margrétarvatn, and Högnavatn), and F) mean monthly NAO (NOAA National Centers for Environmental Information, 2023). For E, NAO annual means are denoted by dashed horizontal lines (red = positive, blue = negative).**



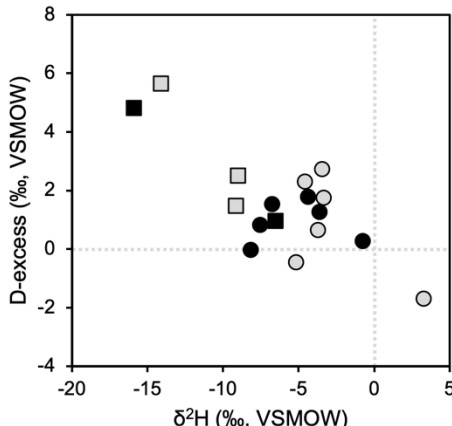

**Figure 5: Difference of September 2020 and 2019 lake surface water δ²H (‰, VSMOW) vs d-excess (‰, VSMOW) from 17 lakes. Squares denote lakes located on the NW highlands (Fig. 1), black and gray fill denote lakes with residence times >1 and <1 year, respectively.**





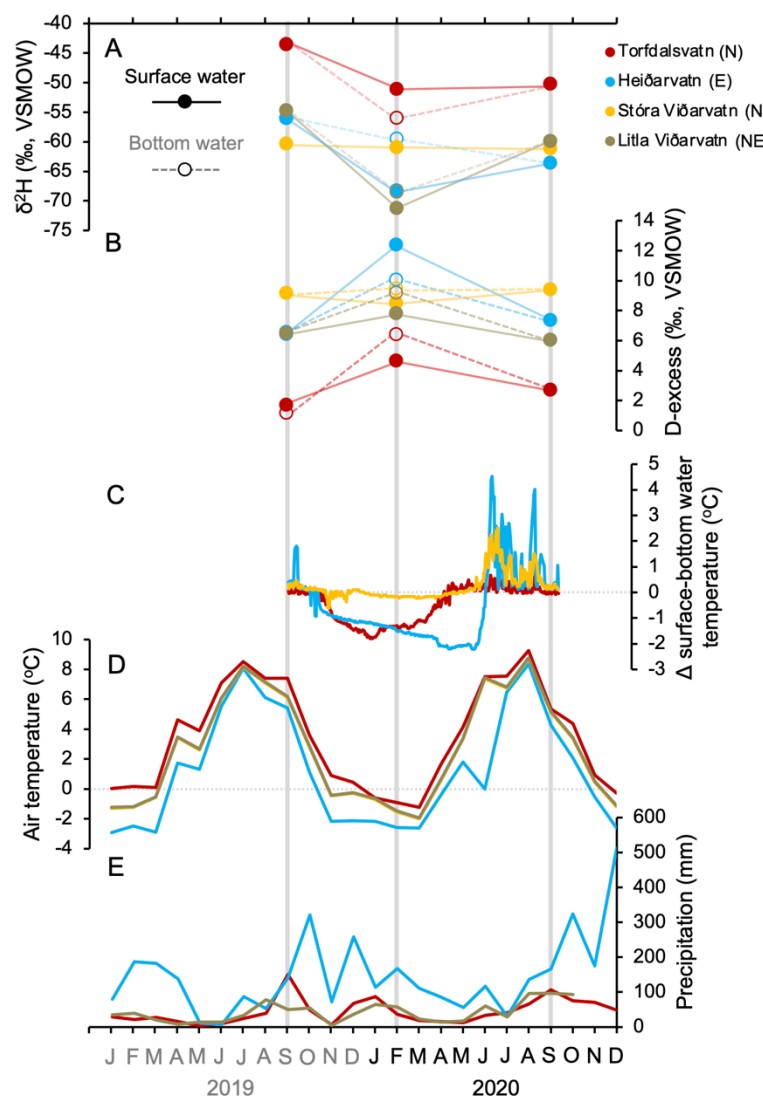

**Figure 6: Interannual variability of lake surface and bottom water δ²H compared with climatological datasets. A) δ²H (‰, VSMOW) of surface and bottom water from 4 lakes collected in September 2019, February 2020, and September 2020, B) corresponding d-excess values (‰, VSMOW), C) difference of *in situ* lake surface and bottom water temperature (ºC) (not available for Litla Viðarvatn, Raberg et al., 2021b), D) mean monthly air temperature (ºC) for each lake, and E) mean monthly precipitation (mm) for each lake (note Litla Viðarvatn is the same as Stóra Viðarvatn).**

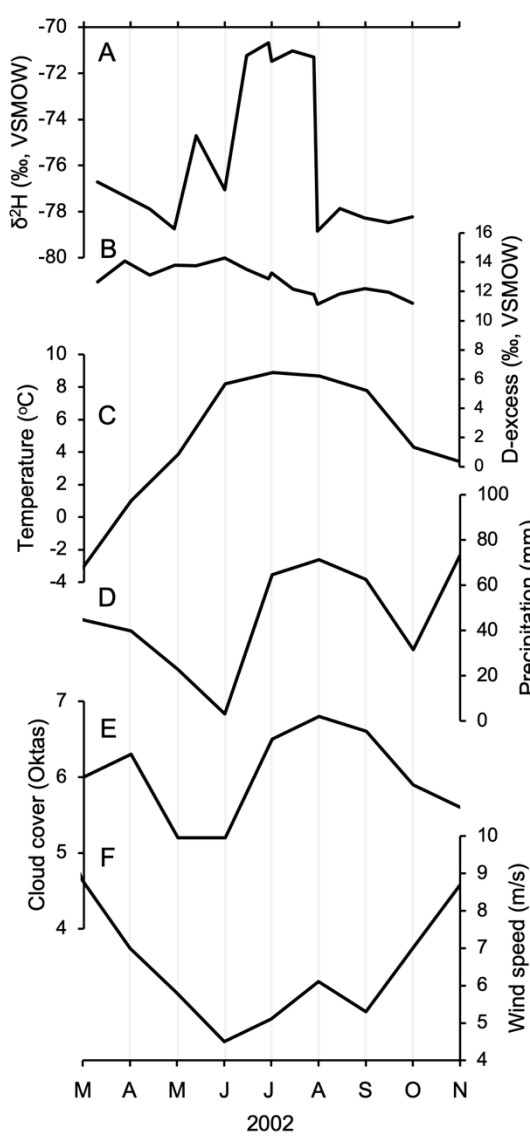

**Figure 7: Timeseries of A) stream outflow δ²H (‰, VSMOW) and B) d-excess (‰, VSMOW) from Efstadalsvatn between March and October 2002 in comparison to local instrumental weather data: C) air temperature (ºC), D) precipitation (mm), E) cloud cover (Oktas), and F) wind speed (m/s).**

δ²H of surface soil water (δ²H$_{soil}$) values from 2019 range from -91.34 to -31.71 ‰ and fall within the natural range of meteoric water isotopes recorded at the Reykjavík GNIP station (Fig. 8). Due to the small size of the δ²H$_{soil}$ dataset compared to lakes, it is difficult to assess if there is any clear spatial variability (Fig. S4). Soil water d-excess values range from -5.04 to 9.84 ‰ (Fig. 8B). In general, surface soil water isotopes fall below the LMWL, although samples collected from beneath densely vegetated moss or grass cover plot along the LMWL (Fig. 8). Five of the sites have 2 soil samples from the same lake





catchment, with the difference of $\delta^2H_{soil}$ values ranging from 1.11 to 21.65 ‰. In general, surface soil waters are $^2$H-enriched

compared to surface lake waters from the same year, although 4 soil water samples are $^2$H-depleted compared to surface lake

water at the location. For temperature, we find a moderate correlation between $\delta^2H_{soil}$ and *in situ* soil mean summer temperature

(r = 0.51, $p < 0.05$) and no significant correlations between mean annual, mean winter, mean autumn, and mean spring

temperature and $\delta^2H_{soil}$ values (Fig. 9). For precipitation, we find a moderate correlation between $\delta^2H_{soil}$ and mean summer

precipitation amount (r = 0.56, $p < 0.05$) and no significant correlation between mean annual, mean winter, mean autumn, and

mean spring precipitation amount and $\delta^2H_{soil}$ values (Fig. 9). In contrast, soil water d-excess is positively correlated with mean

annual (r = 0.58, $p < 0.05$), mean winter (r = 0.61, $p < 0.05$), mean autumn (r = 0.56, $p < 0.05$), and mean spring precipitation

amount (r = 0.50, $p < 0.05$), but not mean summer precipitation amount (Fig. 9). We also find weak to moderate correlations

between d-excess and mean annual (r = 0.58, $p < 0.05$), mean autumn (r = 0.57, $p < 0.05$), mean spring (r = 0.50, $p < 0.05$), and

mean summer precipitation amount (r = 0.33, $p < 0.05$) (Fig. 9). We find moderate correlations between $\delta^2H_{soil}$ and elevation

(r = -0.43, $p < 0.05$) and d-excess and elevation (r = 0.43, $p < 0.05$) and no significant correlation with distance from coast

(Fig. 9). Finally, we find no significant correlation between $\delta^2H_{soil}$ and soil water content (Fig. 9).

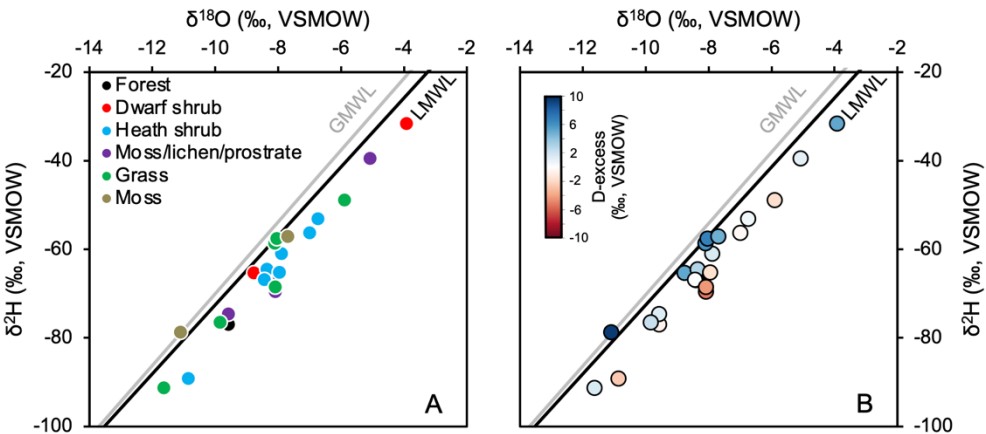

**Figure 8: Soil water stable isotopes (*n*=23) plotted against the GMWL and LMWL. A) samples colored according to the overlying vegetation as described during collection in September 2019 and B) samples colored by d-excess values (‰, VSMOW).**






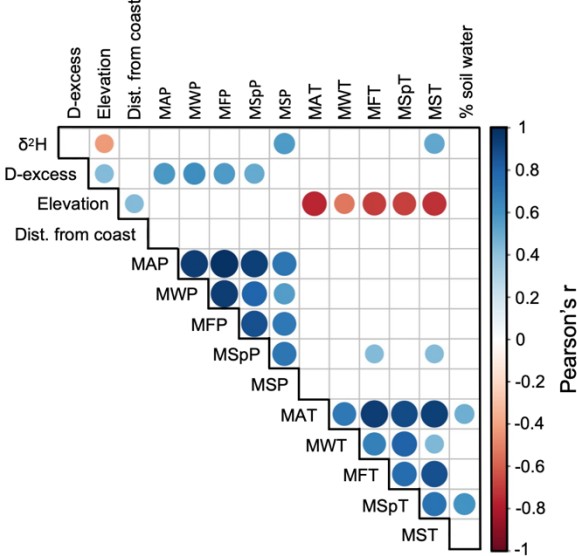

**Figure 9: Pearson correlation matrix of soil water δ²H (‰, VSMOW), d-excess, and various climatological, geographic, and environmental parameters (*n*=23). Empty boxes reflect statistically insignificant correlations (*p* > 0.05).**

Student *t*-tests for lake and soil water populations show that δ²H and δ¹⁸O values are not statistically different between the two groups (*p* > 0.05). However, student t-tests for d-excess, which takes in account both δ²H and δ¹⁸O variables, show that lake and soil water populations are statistically different from one another (*p* < 0.05). This supports the separate assessment of lake and surface soil water isotopes in Iceland.

## 5 Discussion

### 5.1 Lake water isotopes

#### 5.1.1 Environmental controls

Lake water isotopes in Iceland generally plot close to or on the LMWL (estimated using GNIP data from Reykjavík, IAEA/WMO, 2015), demonstrating that most lake water in Iceland is isotopically consistent with unmodified precipitation and a low temperature/high precipitation environment. Lakes with δ²H$_{lake}$ values that generally plot on the LMWL are all open lakes (Fig. 2A) with active inflows providing recharge from meteoric water that falls in the catchment and is stored according to lake residence times (Jonsson et al., 2009). The strongest predictor of δ²H$_{lake}$ in open lakes is cold season temperatures: winter (r = 0.53, *p* < 0.05), autumn (r = 0.62, *p* < 0.05), and spring (r = 0.63, *p* < 0.05) (Fig. 3B). However, due to the lack of correlation between GNIP δ²H values of precipitation and air temperature (Fig. S1A), we do not interpret δ²H$_{lake}$ as a function



of temperature, but rather precipitation sourced from cool seasons. This is supported by the fact that open lakes with residence times of less than one year (efficient flushing) plot close to the LMWL (Fig. 2C), which was found to be biased towards the cold season in Reykjavík GNIP precipitation data (Fig. S1B) and the fact that a larger proportion of precipitation is delivered in autumn/winter (Fig. 1C). Surprisingly, we observe no relationship between precipitation amount and $\delta^2H_{lake}$ (Fig. 3B), which may be due to the large standard deviation of seasonal precipitation amounts recorded at the Reykjavík GNIP station (Fig. 1C).

Open lakes that have residence times of greater than one year (less efficient flushing) deviate from the LMWL more than lakes with residence times less than one year (Figs. 2C and S1). Following the seasonal bias of LMWLs (Fig. S1B), this indicates that the dominant proportion of precipitation falling in these lake catchments falls in the cold season (plots close to annual LMWL) as well as possibly summer (plots below annual LMWL). For the latter "summer" lakes, the low corresponding d-excess values imply there is some degree of seasonal evaporation (Fig. 2D), meaning that precipitation may still be dominantly sourced from cold seasons, but that repeated episodes of evaporation may obscure the seasonal signal in lakes with longer residence times.

Several other mechanisms also help to explain the trends observed in Icelandic lake water isotopes. First, 49 lakes in our study are closed basins meaning that the meteoric water that falls in the catchment is contained until it is lost through evaporation. In Iceland, the effect of evaporation (kinetic fractionation) is observed through the deviation of closed lakes' water isotope values along the LEL and relatively low d-excess values (Fig. 2D). Given the maritime and humid climate of Iceland, and that some of the evaporative lakes are located on the coast, this observation contrasts with documented lake water isotopes in the higher-latitude Arctic where evaporation is minimal in coastal lakes (Leng and Anderson, 2003; Cluett and Thomas, 2020; Kjellman et al., 2022). Compared to these other Arctic lake studies, we hypothesize that the relatively warmer temperature, low amounts of summer precipitation and high wind speeds in Iceland, particularly along the coast and in the highlands (Einarsson, 1984), may contribute to enhanced seasonal evaporation during the relatively longer open water season in these coastal closed lake basins (e.g., Feng et al., 2016; Gonfiantini et al., 2020). In addition, as winds are typically strongest in winter (Einarsson, 1984), a partially ice-free cool season, when the water-air temperature difference is greatest, may contribute to enhanced lake water evaporation as well. Second, we observe moderate and weak correlations between $\delta^2H_{lake}$ in all lakes and two geographic parameters, respectively: elevation (r = -0.46, $p < 0.05$) and distance from coastline (r = -0.33, $p < 0.05$) (Fig. 3A). Icelandic lakes that are farther inland tend to be at higher elevation, thus Rayleigh distillation is also a control on the spatial variability of lake water isotopes in Iceland, consistent with other surface water, near-surface groundwater, and precipitation stable isotopes studies (Bödvarsson, 1962; Friedman et al., 1963; Árnason, 1976; Stefánsson et al., 2019).

### 5.1.2 Interannual variability

Seven lakes in Iceland were sampled for surface lake water isotopes during September 2014, 2019, and 2020, and demonstrate interannual $\delta^2H_{lake}$ variability ranging from 0.24 (Margrétarvatn, NW Iceland) to 17.88 ‰ (Heiðarvatn, E Iceland) (Fig. 4A) and d-excess variability ranging from 1.69 (Torfdalsvatn, N Iceland) to 10.83 (Högnavatn, NW Iceland). In all seven lakes, $\delta^2H_{lake}$ values in 2019 are relatively enriched and have lower d-excess values than from 2014 and 2020. For some lakes, winter





snowfall may be a factor explaining $\delta^2H_{lake}$ values, particularly for those at higher elevations. At Heiðarvatn (442 m asl, E Iceland), $^2$H-depleted snow from the preceding winter may have yielded a winter-biased signal that lingered into the summer

season for some years. We note that the winter preceding September 2014 featured substantially higher amounts of winter snowfall (1063 mm) than the winter preceding September 2019 (553 mm) at Heiðarvatn (Fig. 4D), consistent with the relatively $^2$H-depleted values in 2014 (17.88 ‰, Fig. 4A). For the low-elevation, coastal lake Torfdalsvatn (N Iceland), the amount of winter precipitation did not substantially vary between 2014 and 2020 as it did for Heiðarvatn. However, local relative humidity at Torfdalsvatn was significantly lower through the winter, spring, and summer in 2019 compared to 2014 and 2020 (Fig. 4E).

As Torfdalsvatn is a closed lake, the relative enrichment of $\delta^2H_{lake}$ values and lower d-excess values in 2019 (Fig. 4A-B) are consistent with relatively dryer conditions during the same year.

Seventeen lakes were sampled in both September 2019 and 2020, and reveal interannual variability, with $\delta^2H_{lake}$ value differences between 2019 and 2020 ranging from -3.31 (Skjaldarvatn, W Iceland) to 15.89 ‰ (Margrétarvatn, NW Iceland) and d-excess value differences ranging from -1.70 (Skjaldarvatn, W Iceland) to 5.65 ‰ (Berufjarðarvatn, NW Iceland) (Fig.

5). In general, we observe lower $\delta^2H_{lake}$ values and higher d-excess values in 2020 compared to 2019, except for one lake (Skjaldarvatn), which has higher $\delta^2H_{lake}$ values and lower d-excess values in 2020 (Fig. 5). We predicted that lakes with residence times of less than one year (efficient flushing) would feature larger interannual variability (e.g., Gibson et al., 2002; Leng and Anderson, 2003). However, lake residence time does not correlate with the magnitude of $\delta^2H_{lake}$ value difference between 2019 and 2020 (Fig. 5), consistent with our observations in the total lake dataset (Fig. 2C). However, we do observe

greater $\delta^2H_{lake}$ and d-excess difference for lakes in the NW highlands (squares) compared to lakes elsewhere in Iceland (circles, Fig. 5). We note that during the 2020 sample collection there was fresh snow accumulation in the catchments of the NW highland lakes in September whereas we did not observe this at the same time in 2019 (Fig. S5). This is consistent with more $^2$H-depleted values in 2020 relative to 2019 and thus snowmelt at high elevations may partially explain the relatively $^2$H-depleted values for NW highland lakes compared to lower elevation sites elsewhere in Iceland.

Finally, given that lake water isotope values generally reflect unmodified precipitation (Fig. 2), precipitation source also likely impacts the Icelandic $\delta^2H_{lake}$ values. Such an effect is already apparent in the $\delta^2H$ values of precipitation in Reykjavík, which show a negative correlation with the NAO index (Baldini et al., 2008). Variation of the NAO will impact the strength and location of westerly winds and storm tracks on interannual to decadal cycles (Hurrell et al., 2003; Olsen et al., 2012), where NAO+ results in stronger northerly sources ($^2$H-depleted, cool, and dry) and NAO- results in more southerly

sources ($^2$H-enriched, warm, and wet) to Iceland, which can differ by up to 100 ‰ (Bowen and Revenaugh, 2003; Hurrell et al., 2003). In 2019, the annual NAO index was negative (-0.32), whereas 2014 and 2020 were positive (0.19 and 0.29, respectively, Fig. 4F, NOAA National Centers for Environmental Information, 2023). Our dataset demonstrates more $^2$H-enriched water isotopes with lower d-excess values in 2019 relative to 2014 and 2020 (Fig. 4A-B), which is consistent with the NAO index and the associated changes in moisture source and relatively humidity (Fig. 4F). Hence, precipitation source

is likely a contributing influence on $\delta^2H_{lake}$ values across Iceland on at least interannual timescales.





### 5.1.3 Seasonal variability

Four lakes were sampled for surface lake water three times in two different seasons between 2019 and 2020: September 2019, February 2020, and September 2020. In addition, we took bottom water samples from two lakes in September 2019 (Torfdalsvatn and Heiðarvatn) and all four lakes in February 2020 (Torfdalsvatn, 4.4 m depth, Heiðarvatn, 13 m depth, Stóra Viðarvatn, 19.8 m depth, and Litla Viðarvatn, 1.5 m depth) (Fig. 6A). In general, we observe more depleted surface $\delta^2H_{lake}$ values in winter, which likely reflects the seasonal change in precipitation, where winter snow is isotopically depleted relative to rain that dominates the other seasons (Fig. 6A). In addition, winter d-excess values of surface water are higher than in summer, except in Stóra Viðarvatn, which may reflect varying degrees of summer evaporation in closed (Torfdalsvatn) and even open lakes (Heiðarvatn and Litla Viðarvatn). Compared to surface lake water, winter bottom $\delta^2H_{lake}$ values are relatively depleted in Torfdalsvatn, relatively enriched in Heiðarvatn and Litla Viðarvatn, and equivalent in Stóra Viðarvatn (Fig. 6A). Winter bottom water d-excess values are relative higher in Torfdalsvatn, Litla Viðarvatn, and lower in Heiðarvatn, and similar in Stóra Viðarvatn (Fig. 6B). While the difference varies between lakes, the general trends likely reflect the winter stratification of the water columns. Using *in situ* daily lake water temperatures from the same depths that $\delta^2H_{lake}$ and d-excess values are from for three of the lakes (Torfdalsvatn, Heiðarvatn, and Stóra Viðarvatn, Fig. 6C, Raberg et al., 2021b), we observe strong winter water column stratification in Torfdalsvatn and Heiðarvatn but not in Stóra Viðarvatn, consistent with differences in surface and bottom $\delta^2H_{lake}$ and d-excess values, where Torfdalsvatn and Heiðarvatn's $\delta^2H_{lake}$ values differ by 4.92 and 8.62 ‰, respectively, and Stóra Viðarvatn only by 0.06 ‰ (Fig. 6A). For Litla Viðarvatn, in the absence of continuous bottom water temperature measurements, Sonde water quality measurements (i.e., lake water temperature, pH, dissolved oxygen and conductivity) from September 2019 and February 2020 (Raberg et al., 2023) demonstrate that this lake is also stratified during the winter (Fig. S6), consistent with a difference between surface and bottom water $\delta^2H_{lake}$ values of 2.74 ‰ (Fig. 6A).

In addition to seasonal $\delta^2H_{lake}$ values, we also collected bimonthly water samples from the outflow of a low-elevation lake in northwest Iceland (Efstadalsvatn, Fig. 1A) between March and November 2002. Efstadalsvatn has a short residence time of ~0.1 year (Table 1), meaning that meteoric water should be efficiently recharged through the lake. Compared to the seasonal changes of precipitation $\delta^2H$ values in Reykjavík (Fig. 1D), the $\delta^2H$ values of the outflow water are relatively dampened, ranging from -78.75 ‰ at the end of April to -70.68 ‰ at the end of June (Fig. 7A), but similar in scale to the interannual and seasonal variability observed in $\delta^2H_{lake}$ values from other lakes (Figs. 4-6). Notably, we observe relatively [2]H-depleted values between March and April, followed by rapid [2]H-enrichment of 5.83 ‰ at the beginning of June, and then rapid [2]H-depletion of 7.56 ‰ at the end of August after which $\delta^2H$ values are stable and low (Fig. 7A). The relatively [2]H-depleted values in March and April may be due to lingering snowmelt in the catchment associated with relatively cool seasonal air temperatures (Fig. 7C). The [2]H-enrichment observed at the beginning of June occurs at the same time as an increase in precipitation amount (Fig. 7D), which is likely in the form of rain (as opposed to [2]H-depleted snow) due to above freezing air temperatures (Fig. 7C). Evaporative enrichment does not likely explain this step change as Efstadalsvatn has an open catchment and short residence time and d-excess values are high and inconsistent with a kinetic fractionation effect (Fig. 7B). The cause





of the rapid $^2$H-depletion at the end of August is, however, enigmatic. Air temperatures remain above freezing from the end of
March through September (Fig. 7C), meaning that a change to $^2$H-depleted snow cannot account for the change in $\delta^2$H values.
The other local weather variables also show no changes at this time that could force a change in the $\delta^2$H values (Fig. 7D-F).
While there is no data for Efstadalsvatn, surface and bottom water isotopes and temperatures from Torfdalsvatn, a similarly
shallow lake (3.8 m vs 5.8 m depth, respectively, Caseldine et al., 2003), show that it is dimictic and seasonally stratified from
early April through October (Fig. 6A-C). This suggests that Efstadalsvatn could also be seasonally stratified resulting in
summer rainfall influx flushing through the lake without mixing with the deeper water (e.g., Boehrer and Schultze, 2008).

**5.2 Surface soil water isotopes**

Isotopes of surface soil water collected in summer in Iceland correlate significantly with mean summer environmental
parameters, i.e., precipitation amount (r = 0.56, $p < 0.05$) and temperature (r = 0.52, $p < 0.05$) (Fig. 9), suggesting that summer
is the dominant source water season. In contrast, d-excess values correlate with mean annual (r = 0.58, $p < 0.05$), mean winter
(r = 0.61, $p < 0.05$), mean autumn (r = 0.56, $p < 0.05$), and mean spring precipitation amount (r = 0.50, $p < 0.05$) (Fig. 9). As
most Icelandic surface soil water isotopes plot below the LMWL and have low to negative d-excess values (Fig. 8), the pattern
of soil waters isotopes is consistent with kinetic fractionation and $^2$H-enrichment. Despite summer months typically being
drier in Iceland (Fig. 1C), the seasonal correlations of d-excess with winter, spring, and autumn precipitation, suggest that
evaporation can occur throughout the year. While the residence time of Icelandic soil water is unknown, the residence time of
soil water in other wet, low-energy North Atlantic locations is typically controlled by soil type and overlying vegetation
(Tetzlaff et al., 2014; Geris et al., 2015a, b; Sprenger et al., 2017). The andic properties of the soil types in Iceland (i.e., histosol
and brown-gleyic andosol) result in relatively high water retention (Arnalds, 2004), consistent with the correlations of
precipitation and d-excess values throughout the year. As soil temperatures at these sites periodically dip below freezing during
winter months (Fig. S7), seasonal freezing of the ground may contribute to the warm season precipitation bias observed in
$\delta^2$H$_{soil}$ values by preventing the incorporation of cold season water into a frozen water-soil matrix. When soil samples are
separated by the type of vegetation cover, we observe that moss- and grass-covered soil samples associated with marsh
environments plot closer to the LMWL and have higher d-excess values compared to other types of vegetation cover (Fig. 8).
This likely reflects the saturated marsh environment that these soils were sampled from, which have high water tables, retain
water for longer than soils with efficient drainage, and are less prone to evaporation.
For 5 locations, we took 2 surface soil samples from the same lake catchments, but different locations, also measured
for lake water isotopes (Heiðarvatn, Másvatn, Stóra Viðarvatn, Systravatn and Torfdalsvatn). The $\delta^2$H$_{soil}$ value differences for
the samples from the same lake catchments range from 1.11 (Systravatn) to 21.65 ‰ (Heiðarvatn) and corresponding d-excess
value differences range from 0.54 (Systravatn) to 5.66 (Heiðarvatn). The high variability observed in $\delta^2$H$_{soil}$ and d-excess
values from the same location suggests that additional processes beyond climate and geography contribute to soil water isotopic
variability in Iceland. This heterogeneity may be one reason why correlations are weaker between $\delta^2$H$_{soil}$ values and mean
summer temperature (r = 0.52, $p < 0.05$), precipitation amount (r = 0.52, $p < 0.05$), and elevation (r = -0.43) (Fig. 9) compared





to the same correlations found with $\delta^2H_{lake}$ values (Fig. 3). While vegetation cover may explain some spatial variability in the broader Icelandic dataset, it does not likely account for intersite variability. For example, the sites with high intersample $\delta^2H_{soil}$ value differences (Heiðarvatn = 21.65 ‰ and Stóra Viðarvatn = 11.37 ‰) are both covered by the same vegetation type (moss and heath shrub, respectively), while the only location that had different vegetation cover (Másvatn, grass and heath shrub) had one of the lowest $\delta^2H_{soil}$ value differences (2.04 ‰). Variations in snow cover duration may be another possible explanation for inter-site variability. However, soil temperatures (3-hour resolution) for the following year show that sites with larger $\delta^2H_{soil}$ value differences (e.g., Heiðarvatn and Stóra Viðarvatn) have melt out differences of 3 and 8 days, respectively, whereas sites with small $\delta^2H_{soil}$ value differences (e.g., Másvatn, Systravatn and Torfdalsvatn) show melt out differences with a larger range from 3 days to 13 days (Fig. S7). Therefore, while *surface* soil water isotopes in Iceland may broadly reflect summer to annual precipitation, local climate parameters, fine-scale observations of vegetation cover, and *in situ* thermal regimes do not explain their site-to-site variability.

**5.3 Implications for paleoclimate proxies**

Lake water isotope proxies have become an increasingly common tool to infer past changes in the high-latitude hydrologic cycle, including seasonality of precipitation (e.g., Kjellman et al., 2020; Thomas et al., 2020; Corcoran et al., 2021), moisture balance (e.g., Anderson and Leng, 2004; Balascio et al., 2018), and precipitation source (e.g., Cowling et al., 2022; van der Bilt et al., 2022). Foundational to any paleohydrological study is a clear understanding of the local environmental controls behind water isotope variability. For Icelandic lake water, we find that cold season precipitation, lingering snowpack in fringe seasons, summer relative humidity, and NAO cyclicity are important modulators of $\delta^2H_{lake}$ on interannual timescales, particularly for lakes with sub annual residence times. These are important considerations as they indicate that the seasonality of runoff may not always align with the seasonality of precipitation. Combined with the differences we observe between surface and bottom water $\delta^2H_{lake}$, the complexities of $\delta^2H_{lake}$ are consistent with the poorer relationships observed between typically aquatic mid-chain plant waxes and $\delta^2H_{lake}$ values vs. typically terrestrial long-chain plant waxes and $\delta^2H$ of precipitation values (McFarlin et al., 2019). Surface soil water isotopes generally support previous assumptions that terrestrial biomarker water isotopes reflect summer precipitation (e.g., Balascio et al., 2013; Curtin et al., 2019; Kjellman et al., 2020; Thomas et al., 2020). However, we stress that these soil water isotopes are from the *surface*, and evidence suggests that high-latitude plant roots may source water from the surface (Amin et al., 2020) as well as from below the surface layer (e.g., Eensalu et al., 2023). So, while the seasonal bias may be similar deeper in the soil profile, the enrichment observed in Icelandic surface soils is not likely incorporated in terrestrial plant growth water if sourced from deeper soils. Hence, while still valuable, this soil data currently provide a limited perspective on soil water isotope systematics needed to inform plant growth and terrestrial leaf wax water isotopes studies in Iceland.

In Iceland, there are only two Holocene records of paleoprecipitation developed from water isotopes proxies, one using aquatic invertabrates (i.e., chironomids) from a lake in northeast Iceland (Stóra Viðarvatn, Wooller et al., 2008) and another using leaf wax n-alkanes from a fjord in northwest Iceland (Moossen et al., 2015). Both records observe inconsistencies





with known temperature variability, and through multi-proxy comparisons, infer that the changes in water isotope were related to precipitation. The dominant controls that we find in modern lake water isotopes support prior assumptions that chironomid $\delta^{18}O$ values from Stóra Viðarvatn encode information about past changes in precipitation seasonality and moisture source (Wooller et al., 2008). However, we do note that Stóra Viðarvatn has one of the longest residence times in the dataset (~17 years, Table 1) and the lake's modern water isotopes appear insensitive to interannual climate variability (Fig. 6). While

residence time is not a strong predictor of lake water isotope variability in Iceland, the relative insensitivity of Stóra Viðarvatn to annual changes in moisture supply likely results in a decadally-averaged paleohydrological record. On the other hand, the fjord record from northwest Iceland is based on terrestrial leaf waxes that were subsequently transported to the fjord (Moossen et al., 2015), which are complicated by unknown transport and residence times as well as uncertainties in marine age model reservoir corrections. The relatively limited perspective that our modern surface soil water isotopes provide on terrestrial

biomarker water isotope proxies also makes it challenging to inform the fjord record, although it remains likely that the fjord leaf wax $\delta^{2}H$ values are summer biased. Looking forward, the strong linear relationship between lake water and aquatic invertebrates (e.g., chironomids) oxygen isotopes (Verbruggen et al., 2011) suggests their application in Icelandic lake sediment records is promising. However, further calibration of terrestrial and aquatic plant waxes, their isotopic variability, relationship to source waters, and relative contribution to the sedimentary pool is vital to further advance the use of biomarker-

derived water isotope proxies in Icelandic sedimentary records (e.g., McFarlin et al., 2019; Dion-Kirchner et al., 2020; Hollister et al., 2022).

       Our modern lake and soil water isotope dataset provides important insight into the impact that various geographic, geomorphic, and environmental variables have on the distribution of water isotopes across Iceland. Importantly, the data open the possibility for optimized site selection and more informed paleohydrological studies in Iceland. For example, we identify

lakes that are sensitive to evaporative enrichment, whose sediments can be useful in reconstructing Holocene moisture balance and lakes that record unmodified meteoric water, which offer the possibility of reconstructing changes in precipitation seasonality and moisture source related to atmospheric patterns. Collectively, these paleohydrological records can help address open questions related to, for example, the impact of precipitation on the Holocene mass balance of Icelandic ice caps and shifts in terrestrial plant communities. As both glaciers and terrestrial plants are part of positive feedback loops in the climate

system through changes in albedo and evapotranspiration (Serreze and Barry, 2011), understanding the paleohydrology of Iceland is critical to better understand the past evolution of climate, glaciers and ecology and predict future climate and environmental scenarios over the 21st Century.





## 6 Conclusions and recommendations

We report Icelandic lake, stream, and soil water isotopes ($\delta^2$H and $\delta^{18}$O) measured from samples collected in 2002, 2003, 2004, 2014, 2019 and 2020. Guided by correlation analyses between $\delta^2$H, d-excess, and a suite of climatological, geomorphic, and geographic parameters, we draw the following conclusions:

- Lake water isotopes reflect local precipitation with biases toward the cold season, particularly for lakes with sub-annual residence times, consistent with the time of year when most precipitation is delivered. A subset of lakes, which are closed basins, demonstrate $\delta^2$H$_{lake}$ and d-excess values affected by evaporative enrichment, which likely occurs during the typically open-water summer season when relative humidity is lower, lake water surface temperatures are elevated, and the lake surface is exposed to wind. Lake water isotopes are generally $^2$H-depleted at higher elevations and at locations farther from the coast, consistent with Rayleigh distillation.

- Interannual variability of lake water isotopes and d-excess values are consistent with observations that 2019 received less winter snowfall, was relatively dry during the summer, and was characterized by a positive NAO, all contributing to $^2$H-enriched and low d-excess lake water relative to 2014 and 2020.

- Seasonal variability of lake water isotopes shows that lake surface water is relatively $^2$H-depleted with higher d-excess values in February compared to September, likely due to the predominance of snow over rain in winter months. Additionally, bottom water isotopes reflect winter stratification that is independently recorded in the seasonal temperature and water chemistry cycles of the lakes.

- Isotopes and d-excess values of summer surface soil water reflect local summer precipitation overprinted by evaporative enrichment that can occur throughout the year. Water isotope measurements from samples collected from just 10s of meters apart can show large variability, suggesting that additional factors beyond the local climate, especially topography and vegetation, which are inherently related, may be important factors.

- The collective dataset optimizes site selection for future studies in Icelandic paleohydrology focusing on changes in moisture balance, precipitation seasonality, and moisture source, and provides a process-based framework to guide downcore water isotope proxy interpretations. Specifically, sedimentary records from closed basin lakes offer the potential to reconstruct Holocene changes in precipitation-evaporation balance while sedimentary records from open basin lakes, particularly those with sub-annual residence times, may provide insight into past changes in moisture source related to winter snowfall variability and atmospheric circulation patterns. At present, the simplest sedimentary records of precipitation would be derived from aquatic invertebrate (e.g., chironomids) oxygen isotopes given the strong linear relationship observed between lake water and aquatic invertebrate oxygen isotopes. Future work is needed to constrain the relationship between aquatic plant waxes, their isotopic variability and relationship to lake water before secure paleohydrological studies can be developed from them. Similarly, given the complexity and



uncertainty observed in Icelandic surface soil water isotopes, we urge caution in the application of terrestrial water isotope proxies, such as long-chain *n*-alkyl lipids, before further calibration studies are conducted.

**Data availability**

Submitted to the Arctic Data Center.

**Competing interest**

The authors declare that they have no competing interests.

**Acknowledgements**

This study has been supported by a GSA Graduate Student Grant and Comer Science and Education Foundation funding (ÁG and YA), the Icelandic Centre for Research (Warm Times-Cold time Grant-of-Excellence #022160-002, ANATILS Grant-of-
Excellence #130775-052, ÁG and GHM; Doctoral Student Grant #163431-053, DJH), the Doctoral Grant of the University of Iceland (CRF), and the National Science Foundation (ARCSS-1836981, GHM, ÁG, and JS; EAR-1928303, SK and JMM). We thank Sædís Ólafsdóttir, Jason Briner, Ragna Aðalsteinsdóttir, Pete Langdon, Ian Holmen, Þorsteinn Jónsson, and Sveinbjörn Steinthorsson for assistance in the field.

**Author contributions**

ÁG, GHM, JS, YA, and SK funded the research; YA, CRF, JHR, DJH, ÁG, GHM, and JS collected lake water and soil samples; KBÓ provided Icelandic meterological data; DJH, JMM and JHR processed and interpreted the data; DJH led the writing of the manuscript with contributions from all co-authors.

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
