# Peer review of "Spatiotemporal variation of modern lake, stream, and soil water isotopes in Iceland"

_Hydrology and Earth System Sciences, 2024_

## Author Response (AR2)

**Reviewer 1**

In this manuscript, Harning et al. present a detailed investigation of Icelandic hydrology through a multi-year isotopic (δ2H, δ18O, d-excess) observations of lake, stream, and soil water. The number of isotopic observations here represents a remarkable dataset that will be a valuable contribution to the Arctic isotopic (modern and paleo) community. The authors explore the isotopic observations across a range of spatial and temporal scales to understand the key physical drivers of isotopic variations of these waters. They strive to place these observations in the context of paleoclimate studies that utilize these isotopic ratios in different archives to investigate past climatic changes. The authors offer a range of important recommendations for paleoclimate researchers through their modern water isotopic observations.

I think this rich dataset and set of analyses are certainly worthy of publication. However, there are some comments that I feel should be addressed before being published.

*We kindly thank the reviewer for their valuable feedback on our manuscript. Below, we address each of their comments with our reply (italicized) and reference to changes made in the manuscript.*

**Major comments**

1. Use of different datasets.

One area that limits the effectiveness of this investigation is uncertainty in which parts of the dataset are being discussed in any given section. There are some times when the authors discuss the full set of data while the majority of time they focus only on those samples in 2014/19/20. It is unclear why there is not a greater focus on the full dataset. My recommendation would be to either treat all data equally throughout the manuscript, or just focus on the 2014/2019/2020 data (and you could include a subsection on the old data to provide additional context if desired). The language used at times is clearly just in thinking about the 14/19/20 data, but it is not written explicitly that this is what the authors are discussing. Some of my minor comments below might simply be related to me mistaking what subset of lakes you are discussing.

As presented, the early years of data are often just an afterthought, where they could actually provide significant value. Especially for the implications to paleo proxies, I do not understand why a closer examination of the earlier data is not completed. Having 6 years of data comparison as compared to 3 years is a big difference for constraining the key drivers of lake water changes. I am guessing you cannot do the repeated comparison of the 7 resampled lakes, but there are plenty of other examinations of the data across all the years that could be completed (see below).

*We appreciate this suggestion and the opportunity to elaborate why we focused on the 2014, 2019 and 2020 datasets. The primary reason is that the 2002, 2003, and 2004 datasets feature different lakes from the 2014-2020 dataset. Additionally, these earlier lakes feature very different sample sizes per a year (2002=20, 2003=148, and 2004=3) and do not have common lakes across all three years like 2014-2020 lakes have that enable the detailed interannual comparison of water isotopes. Another reason that the 2003-2004 lakes were not included in some discussion points is that the lakes were not measured for depth, meaning that important*

*parameters, such as residence time, could not be estimated. This has been clarified in the Section 3.1 Sample Collection (L169-172 and L177-178).*

*However, we note that the major strength in the 2002 and 2003 lake dataset is that they include a substantially larger number of closed lakes compared to the latter 2004 and 2014-2020 years, which enabled a robust estimation of the LEL across all years sampled (e.g., see revised Fig. 2). In the revised manuscript, this point is emphasized in L169-170.*

2. Interannual comparisons and key drivers.

I think the exploration of interannual comparisons is particularly valuable for their paleoclimate implications. However, I think the analysis could be improved in several important ways that would allow the analysis to more effectively convey the conclusions the authors draw here. For certain changes, I think there are potential alternative explanations for why a given isotopic change occurred, where some additional analysis might be able to better eliminate some explanations and lead readers to the particular conclusions the authors have made.

First, I would recommend including the early lake data (2002, 03, 04) to the interannual variability analysis. Obviously, the climate system has changed rather substantially over this time period, so it would be interesting to see if there are significant differences between this earlier data and the more recent data. Exploring these longer term changes would seemingly help to further evaluate a number of the suggested conclusions of key lake water drivers.

*As mentioned in our response to the previous comment, we cannot include interannual variability of the 2002-2004 lakes as we have done for the 2014-2019 lakes as the lakes are not the same each year. However, we have now included annual LEL and LMWL estimates for each year to assess the broader interannual variability of lakes. We do not calculate a LMWL and LEL for the year 2004 because the sample set is only 3 lakes and none of which are closed lake basins needed for the LEL. We also do not calculate a LEL from 2002 because the there is only 1 closed lake basin from that year. These new LMWL and LELs have been added to a revised Fig. 2 (new panels E and F) and substantial text has been added to the Discussion (L419-444). Importantly, these additions do extend our previous discussion on the 2014-2020 samples that indicated that interannual variability existed likely based on changes in relative humidity. This has also been emphasized in the Discussion (L487-488).*

Related to this point, bringing in the earlier data would allow the authors to compare to the precipitation isotopic record from the GNIP station in Reykjavik. The precipitation isotopic composition would vary significantly around Iceland, but having this information of the isotopic inputs to lake water would be particularly useful to connecting to some of the conclusions on seasonal biases. Looking at the interannual variability in the Reykjavik precipitation isotopic record in comparison to the lake water would be helpful in this assessment, rather than just as a broad climatology assessment (e.g., that winter precip is depleted, etc).

*As the reviewer points out, different regions of Iceland have different sources of precipitation. Therefore, in our opinion, directly comparing the 2014-2020 interannual variability of lake water isotopes, which are from lakes distal to Reykjavik, would not be a relevant comparison. However, it is worth exploring the variability of LMWL and LEL calculated for each year as previously suggested by the reviewer. In this sense, we note that we previously calculated seasonal LMWLs from the GNIP dataset in Reykjavik in the original manuscript (Fig. S1B). We*

*have used this information to derive information about annual LMWL variability (as discussed in the following comment) in the revised manuscript.*

A single Local Evaporation Line (LEL) is determined for the whole dataset. I would recommend computing the LEL each year. This is an easy way to summarize the suite of lake data for each year, instead of needing to compare a single lake to itself directly. This would also allow for easy comparison with the 2002-2004 data, even if not the same exact lakes. The system should respond to similar larger-scale variations, so, while not a perfect one, a comparison of these years would be meaningful and provide greater context for the Icelandic hydrologic system. Spatial variations could also be studied in this manner. The dynamics are certainly different in this location, but studies like that by Leng and Anderson (2003) that examines drivers of interannual and spatial LEL variations in West Greenland or Kopec et al. (2018) that examines the LEL changes in response to the NAO could serve as an example of how to examine the full system instead of (or in addition to) a lake-by-lake approach used here.

*We thank the reviewer for this valuable suggestion and the opportunity to expand our analysis of the entire dataset. Using LMWLs calculated for each year, we find that except for 2002, all years have nearly identical LMWLs. The 2002 anomaly is likely due to the fact lake water samples from this year were only collected from Northwest Iceland, compared to other years that incorporate sites from across Iceland. Therefore, this likely reflects the cooler climate inherent to Northwest Iceland, and the greater proportion of snow compared to rain, as winter LMWL typically plot above the LMWL (Fig. S1B). This aspect has been added to the revised manuscript (revised Figure 2E-F and text L419-444).*

Lastly, a common practice in these sorts of lake water isotopic studies is to examine the intersection of the LEL with the Local Meteoric Water Line (LMWL). This intersection is thought to be the average input of the lake, and then that water evaporates along the LEL (e.g., Gibson et al., 2016). Does this LEL-LMWL intersection reflect the precipitation values observed in Reykjavik? Is there significant interannual/seasonal variability in this intersection? This would be helpful in evaluating the relative seasonal influences in this system and how they might change year to year. In particular, a key conclusion here is that these lakes are driven by cold-season precipitation – does this intersection with the LMWL reflect that conclusion? If so, it is good evidence supporting this conclusion. If not, there might be other factors that should be considered here.

*Again, we thank the reviewer for this suggestion and the opportunity to expand our analysis of the entire dataset. Using LELs calculated for each year, we find some subtle differences in the intersection of the LEL and the LMWL, but all annual LELs intersect with modern ranges of Reykjavik precipitation. The variability of LEL-LMWL intersections is consistent with our previous interpretations, e.g., the occurrence of snowfall-heavy winters in 2020 compared to 2019. Therefore, these suggestions bolster our previous conclusions and are a beneficial addition to our revised manuscript (revised Figure 2E-F and text L429-439).*

**Minor comments**

Line 59. Do not need the second 'however' in this paragraph.

*Edited.*

Figure 1. Precipitation. Why not just use the modern window for Reykjavik precipitation (1992-2018)? It is more relevant to the sampling effort here.

*This is a good point and we have edited the figure to only include the more recent interval between 1992 and 2018. See revised Figure 1B-E, which includes air temperature, precipitation, and water isotopes of precipitation.*

Figure 2. Near end of caption – you write "form" and seemingly mean 'from'.

*Edited.*

Figure 2 – what about the closed lakes outside the B-D narrower axes range? There are many closed lakes with higher δ2H and/or δ18O values. I assume they help define the LEL.

*Panels B-D were originally focused on the 2014-2020 lakes, and therefore narrowed the axes to the range of values from those years. However, we agree it is good to include the earlier years from 2002-2004, which have now been edited in panels B and D. As we do not have lake depth measurements to estimate lake volume needed for residence times (RT) for the 2002-2004 lakes, we cannot include the 2002-2004 lakes in panel C. These aspects have been clarified in the figure caption (L135-140).*

Line 140. More info and/or citations would be helpful on this statement. Could cite Steen-Larsen et al. 2015 for their study on the isotopic composition of water vapor in Iceland. I also think exploring more fully would be important for the NAO discussion later in the manuscript.

*We appreciate the suggested reference and have added it to this section (L158), as well as expanded the paragraph to include an introduction to the NAO for later in the Discussion (L158-162).*

Line 150. Why were these 7 lakes chosen to be revisited? I assume a lot of it is for simple logistics purposes (which is very understandable!), but is there further justification that can be offered for why these ones are particularly good for exploring these interannual changes?

*The 7 revisited lakes were indeed chosen partially out of their ease of access. Additionally, as part of a much larger project aimed at calibrating biomarker proxies for temperature, we selected these lakes as they span a large temperature gradient.*

Line 193. How was catchment area measured in Google Earth? There must be numerous assumptions and uncertain decisions that had to be made in this process. Were any groundtruthed?

*We apologize for the mistake. The catchment areas were measured in QGIS using the ÍcelandsDEM (2x2m resolution, Landmælinger Íslands), and not Google Earth. This has been clarified in the text L225.*

Line 204. I think it is fine to just pick a value here to use, but it would seem quite possible to better inform this number. If it is a generally drier year, the value would likely be smaller, and if it is a wet one, it would be larger. A simple experiment to explore the interannual variation

question later in the manuscript would be to try a few different values of this factor to approximate a drier and wetter year.

*We appreciate this suggestion, however, without further information on lake catchment hydrology to better inform these numbers, we prefer to leave the R/P value as 0.5. We have acknowledged in the methods that the R/P values may also change for a single lake between years due to changes in precipitation-evaporation balance (L238).*

*As an aside, we did experiment with the various R/P values for lakes. For most lakes, the residence time using a R/P value of 0.5 was well over one year, and extreme variations in R/P values generally kept the residence time above one year. In this sense, the lakes we identified to have multi-year residence times using the 0.5 R/P value generally have multi-year residence times with extreme R/P ratios. The same was found to be true for sub-annual residence time lakes. Therefore, the lakes with multi-year residence times will generally have long-term averaged responses to precipitation variability whereas sub-annual residence time lakes will generally have increased sensitivity to seasonal changes in precipitation, regardless of the R/P value used.*

Line 230. The lake with a d2H value of -1.35‰ does not fall within the range of meteoric waters? I think you can just delete that part of the sentence.

If you do want to assess how the waters compare with the Reykjavik precip, then it would probably be better to examine the precip during times close to when the lake water was sampled (e.g., precip for the year prior to the lake sample).

*Good point, we have edited the sentence to say "generally" falls within the range of meteoric waters (L270).*

Line 244-245. The precipitation amount at the 'closest weather station' could be rather different than what falls in the lake basin, particularly with elevation differences. There is not necessarily a better way to do this analysis (you could try a reanalysis product, but that is not ideal either), but I would note that there could be significant uncertainty added in this analysis. You can correct for the spatial differences in temperature using the lapse rate as you did here, but this is a harder one to "correct". I say all this because a weaker correlation to precipitation amount than say temperature might not actually be physically meaningful with the uncertainty here.

*We certainly agree with this point and have now acknowledged this uncertainty in the text (L287-288).*

Figure 4E. The sustained low RH for Torfdalsvatn (red) in 2019 seems very unlikely. Could there be instrument issues (or RH calculation issues) here?

*As there is no indication of instrumental issues reported in the metadata for the weather station, we can only trust that the observed change in RH is genuine.*

Figure 7. It would be worth comparing the Reykjavik GNIP precipitation isotopic values to the stream flow. That could be helpful in assessing the strength or lack of an evaporation signal in this water.

*Thank you for the suggestion. We have added Reykjavik GNIP precipitation isotopic values to Figure 7 (gray in panel A) and to the discussion of Efstadalsvatn stream outflow water isotope values (L237-238).*

Line 328-330. I think this is true for open lakes (as said in the next sentence), but I think there are plenty of lakes that plot significantly different than the LMWL.

*This is correct, as we elaborate in the following paragraph for closed lake basins. Importantly, most lakes are open (n=149) compared to closed (n=49). We have clarified the text to reflect these numbers (L381 and L397)*

Line 384. Consistent with what observation?

*We have removed the latter portion of the sentence to improve clarity (L465).*

Lines 394-395. This does not seem correct to me. My understanding of the NAO is that when it is positive (strong low pressure over Iceland), the North Atlantic storm track centers right on Iceland, and would yield high precipitation and humidity (low evaporation). This setup has strong south to north transport. You can see this in any meteorological or reanalysis dataset (i.e., NAO has strong positive correlations between precipitation, humidity, southerly winds across this region). It would seem quite unlikely for the dominant moisture sources during the positive phase to be from the north. The negative phase removes Iceland from the primary storm track (making it drier), on average. If 2019 is the negative NAO year, then it would seem to me the higher d2H and lower d-excess values are showing an expression of the relative dryness (higher evaporation / lower inputs) while the other two years see higher inputs / lower evaporation, on average.

*We apologize for the confusion and thank the reviewer for clarifying this aspect of the paper. Indeed, NAO+ and NAO- conditions result in warm/wet and cool/dry conditions generally over Iceland (e.g., Hurrell et al., 2003). We have edited the manuscript to appropriately describe these NAO conditions as well as re-evaluated the implications for lake water isotopes, which are now consistent with each other (L475-481).*

Line 396. How is the time delineated here? My interpretation in reading this would be that NAO is computed for the calendar year of each of these years (2014, 2019, 2020). To understand the effects on the lake water, you would likely want some time period (e.g., one year) before the sampling took place. Looking quickly at the data, I do not think it would change your NAO values too much, but it would be a more appropriate assessment.

*Indeed, the original annual NAO values were averages of each month of the calendar year in 2014, 2019, 2020. We appreciate the suggestion and have edited the averages to be the 12 months prior to the month of sampling. This has been clarified in the text (L481-482), Figure 4F and Figure 4 caption (L320). The values are similar so our original conclusions remain the same.*

Line 427-428. Say specifically what the time change is associated with the enrichment / depletion.

*Added L535-536.*

Line 539-541. In the Interannual section, you showed the NAO in 2019 was negative (which is correct), not positive as written here. That said, I do agree with the conclusion overall as written here (if the NAO value was correct) – i.e., drier conditions driving the observed enrichment and low d-excess - but this is not consistent with the argument presented in 5.1.2. I think this is how you should argue the point in 5.1.2.

*Thank you for catching this mistake. Following our revision of the discussion in section 5.1.2., the conclusions are now correct as written.*

**References**

Gibson, J. J., Birks, S. J., & Yi, Y. (2016). Stable isotope mass balance of lakes: a contemporary perspective. Quaternary Science Reviews, 131, 316-328. https://doi.org/10.1016/j.quascirev.2015.04.013.

Kopec, B. G., Feng, X., Posmentier, E. S., Chipman, J. W., & Virginia, R. A. (2018). Use of principal component analysis to extract environmental information from lake water isotopic compositions. Limnology and Oceanography, 63(3), 1340-1354. https://doi.org/10.1002/lno.10776.

Leng, M. J., & Anderson, N. J. (2003). Isotopic variation in modern lake waters from western Greenland. The Holocene, 13(4), 605-611. https://doi.org/10.1191/0959683603hl620.

Steen-Larsen, H. C., Sveinbjörnsdottir, A. E., Jonsson, T., Ritter, F., Bonne, J. L., Masson-Delmotte, V., et al. (2015). Moisture sources and synoptic to seasonal variability of North Atlantic water vapor isotopic composition. Journal of Geophysical Research: Atmospheres, 120(12), 5757-5774.https://doi.org/10.1002/2015JD023234.

**Reviewer 2**

**General Comment**

In this manuscript, Harning et al. undertake a comprehensive study of Icelandic hydrology using multi-year isotopic observations ($\delta 2H$, $\delta 18O$, d-excess) of lake, river and bottom waters. Based on a large number of isotope observations they can draw on a significant data set that will be of great use for Arctic isotope research. The authors represent isotopic variations across different spatial and temporal scales to identify the physical drivers of these variations. They also use different archives to investigate historical climate change. Additionally, based on their modern water isotope data the authors make several important recommendations for paleoclimate researchers.

This study, which is based on very extensive data and places it extremely well in the context of climate change, is definitely worth publishing. Before that, however, I would like to make a few minor comments that should be seen as supplementary to the comments of the first expert and are therefore somewhat smaller in scope.

*We kindly thank the reviewer for their valuable feedback on our manuscript. Below, we address each of their comments with our reply (italicized) and reference to changes made in the manuscript.*

**Minor comments**

Table 1: In the row at site 13, the row in the 2nd column may have slipped.

*Thank you for the comment. We double checked Table 1, and everything appears as we intended.*

Figure 2: I guess you mean from instead of form: „…lakes only form 2014, 2019, and 2020 (n=80)."

*Edited.*

Line 149: Did the farmer followed your sampling protocol?

*Yes.*

Line 165: Is this procedure a standard protocol for the analysis of stable isotopes in soil samples?

*Yes.*

Line 193: Are there major variations of the lake surface area during the year which can have an effect on the calculation of the surface area?

*This is a good consideration, although we are not aware of any significant lake surface area changes during the year.*

Line 384: What observation do you mean?

*We were referring to the observation that lake residence time does not correlate with the magnitude $\delta^2H_{lake}$ value difference between 2019 and 2020, as stated in the sentence. We have removed the latter portion of the sentence to improve clarity (L465).*

Line 427: What time change is associated with the increase/decrease?

*These changes occur over the course of days to weeks, which have now been clarified in the text (L535-536).*